# The Impact of Serum Levels of Reactive Oxygen and Nitrogen Species on the Disease Severity of COVID-19

**DOI:** 10.3390/ijms24108973

**Published:** 2023-05-18

**Authors:** Sameh A. Ahmed, Yaser M. Alahmadi, Yasser A. Abdou

**Affiliations:** 1Department of Pharmacognosy and Pharmaceutical Chemistry, College of Pharmacy, Taibah University, Al Madinah Al Munawarah 30001, Saudi Arabia; 2Department of Clinical and Hospital Pharmacy, College of Pharmacy, Taibah University, Al Madinah Al Munawarah 30001, Saudi Arabia; yahmadi@taibahu.edu.sa; 3Ohud Hospital, Al Madinah Al Munawarah 42354, Saudi Arabia; abo_ali600@hotmail.com

**Keywords:** COVID-19, reactive oxygen species, reactive nitrogen species, tumor necrosis factor-alpha, interleukin-6, angiotensin-converting enzyme 2

## Abstract

Elucidation of the redox pathways in severe coronavirus disease 2019 (COVID-19) might aid in the treatment and management of the disease. However, the roles of individual reactive oxygen species (ROS) and individual reactive nitrogen species (RNS) in COVID-19 severity have not been studied to date. The main objective of this research was to assess the levels of individual ROS and RNS in the sera of COVID-19 patients. The roles of individual ROS and RNS in COVID-19 severity and their usefulness as potential disease severity biomarkers were also clarified for the first time. The current case-control study enrolled 110 COVID-19-positive patients and 50 healthy controls of both genders. The serum levels of three individual RNS (nitric oxide (*NO^•^*), nitrogen dioxide (*ONO^−^*), and peroxynitrite (*ONOO^−^*)) and four ROS (superoxide anion (*O_2_^•−^*), hydroxyl radical (*^•^OH*), singlet oxygen (*^1^O_2_*), and hydrogen peroxide (*H_2_O_2_*)) were measured. All subjects underwent thorough clinical and routine laboratory evaluations. The main biochemical markers for disease severity were measured and correlated with the ROS and RNS levels, and they included tumor necrosis factor-alpha (TNF-alpha), interleukin-6 (IL-6), the neutrophil-to-lymphocyte ratio (NLR), and angiotensin-converting enzyme 2 (ACE2). The results indicated that the serum levels of individual ROS and RNS were significantly higher in COVID-19 patients than in healthy subjects. The correlations between the serum levels of ROS and RNS and the biochemical markers ranged from moderate to very strongly positive. Moreover, significantly elevated serum levels of ROS and RNS were observed in intensive care unit (ICU) patients compared with non-ICU patients. Thus, ROS and RNS concentrations in serum can be used as biomarkers to track the prognosis of COVID-19. This investigation demonstrated that oxidative and nitrative stress play a role in the etiology of COVID-19 and contribute to disease severity; thus, ROS and RNS are probable innovative targets in COVID-19 therapeutics.

## 1. Introduction

Coronavirus disease 2019 (COVID-19) is a global pandemic caused by severe acute respiratory syndrome coronavirus 2 (SARS-CoV-2). Given the tremendous health and economic impacts of the COVID-19 pandemic, all strategies that improve patient conditions, hasten recovery, and lessen the risk of morbidity and mortality are regarded as clinically and economically important [1]. The high mortality rates of COVID-19 have informed explorations of various research avenues to identify potential therapeutic solutions [2]. The severity of coronavirus infections is related to the disruption of the redox equilibrium, which is caused by cumulative oxidative damage, degradation of antioxidative defense mechanisms, and surges in reactive species [3].

Structurally, the four proteins that constitute SARS-CoV-2 are the spike, envelope, membrane, and nucleocapsid proteins. While the nucleocapsid protein protects the RNA genome, the spike, envelope, and membrane proteins collectively build the viral envelope [4]. The spike protein permits the virus to bind to and merge with the cell membrane of a host cell. It is a glycoprotein with two functional components, viz., S1 and S2. The S1 subunit catalyzes attachment and serves as a target for neutralization by antibodies, while the S2 subunit is essential for viral entrance [5]. The S1 subunit also acts as a receptor for angiotensin-converting enzyme 2 (ACE2), inducing the overproduction of reactive oxygen species (ROS) once bound [6]. The generation of ROS, in turn, leads to ROS-dependent cellular signaling, such as activation of the nuclear factor kappa-light-chain-enhancer of activated B cells (NF-κB) pathway, increasing inflammation by enhancing the expression of numerous genes, including tumor necrosis factor (TNF-α), interleukins (IL1-10), chemokines, and colony-stimulating factors, all of which cause endothelial injury and vascular inflammation [7]. The excessive production of ROS causes ROS/antioxidant equilibrium disturbances that lead to the modulation of cellular functions. One of the most significant causes of endothelial cell damage in patients with COVID-19 may be excessive ROS formation in endothelial cells produced by SARS-CoV-2, along with immunological dysregulation (cytokine storm) [8]. Acute respiratory distress syndrome (ARDS), multi-organ failure, and mortality may be associated with this phenomenon, which increases disease severity by activating immunological, endothelial, and platelet cells, thus initiating strong endothelial inflammatory responses [9].

Although several ROS are involved in the pathological processes of COVID-19, hydrogen peroxide (*H_2_O_2_*) is of special relevance since it is highly stable and readily passes through biological membranes. H_2_O_2_ is capable of locally producing the hydroxyl radical (•OH) via an iron-mediated Fenton reaction. Other ROS produced by molecular oxygen include the extremely reactive superoxide anion radical (*O_2_^•−^*) and singlet oxygen (*^1^O_2_*) [10]. Owing to this, the hydroxyl radical, superoxide anion radical, hydrogen peroxide, and singlet oxygen are the major ROS involved in COVID-19-related inflammatory processes.

Reactive nitrogen species (RNS) have been identified as mediators of self-destructive responses in COVID-19 patients, including the nitric oxide radical (*NO^•^*), nitrogen dioxide radical (*ONO^•^*), and peroxynitrite radical (*ONOO^•^*) [11]. Cytokines and endotoxins trigger the inducible isoform of nitric oxide synthetase (NOs), which stimulates the generation of nitric oxide, which then reacts with the superoxide anion to produce the strongly reactive peroxynitrite radical [12]. Local pulmonary and inflammatory cells create RNS (such as nitric oxide) that are involved in the pathophysiology of chronic lung illnesses and pulmonary infections [13]. Highly reactive molecules, such as the hydroxyl radical, nitrogen dioxide radical, and peroxynitrite radical, which break down into the nitrogen dioxide hydroxyl radical, facilitate self-destructive reactions. Cycles of the superoxide anion and nitric oxide radicals play a vital role in maintaining the serum levels of RNS and ROS [14].

Additionally, ROS and RNS are essential for signal transduction. Pattern recognition receptors, interferon response, and interferon regulatory factors are activated by viral proteins or nucleic acids. In addition, they increase the generation and activity of nitric oxide synthase via the myeloid differentiation protein [15].

The key cytokines produced by the immune system in patients with severe COVID-19 are interleukins (IL-1, IL-2, and IL-6), TNF-α, and interferons (IFNs) [16]. IL-1 and IL-2 are well-known activators of ROS and RNS production, while IL-6 stimulates human neutrophils and monocytes, thereby boosting the production of free oxygen radicals [15,16]. TNF-α and IL-6 trigger ROS generation in mitochondria in correlation with ATP synthesis. In addition, individuals with COVID-19 receiving treatment in intensive care units have been reported to have a higher risk of death when their IL-6 levels are elevated [17]. To help manage viral infection, IFN-α, β, and type III are produced and activate genes that have direct antiviral effects. However, interferons stimulate the production of RNS in humans [18].

Furthermore, ACE2, a multifunctional transmembrane protein, serves as a cellular receptor for SARS-CoV-2 spike proteins and is crucial in alleviating COVID-19-induced inflammatory and oxidative injury. The ACE2 enzyme reduces angiotensin II, an NADPH oxidase stimulator, thus reducing superoxide production. Furthermore, angiotensin 1-7, a byproduct of ACE2 enzymatic action, has a high antioxidant impact [19].

Since ROS and RNS have short half-lives, detecting their concentrations is challenging, particularly in biological fluids. However, the development of certain fluorescence-based probes has facilitated such measurements [20]. Thus, a recent study assessed individual ROS and RNS in the sera of rheumatoid arthritis patients [21] and healthy subjects that were smokers [22]. However, most of the research on ROS and RNS in COVID-19 patients has been focused on measuring overall oxidative stress rather than examining individual ROS and RNS and their related pathogenic functions.

To cover some of these gaps, the aim of this research was to assess the levels of individual ROS and RNS in the sera of COVID-19 patients by utilizing selective fluorescence probes and linking/matching the serum concentrations with other COVID-19 laboratory biomarkers. Further, the serum concentrations of the studied reactive species that represent effective markers of COVID-19 severity were elucidated.

## 2. Results

### 2.1. Demographic Features and Laboratory Test Results

This case-control study consisted of two groups: COVID-19 patients (*n* = 110) and healthy controls (*n* = 50) (Table 1).

There were no notable variations in the two groups in terms of body mass index (BMI), height, weight, or age. The CT scans of suspicious patients showed ground-glass opacities; however, a confirmatory diagnostic procedure that involved oropharyngeal swab sample collection and nucleic acid analysis testing was performed. Thirty-six of the COVID-19 patients were admitted to the ICU at Ohud Hospital due to disease severity, while the remaining 74 patients were not admitted (non-ICU patients).

The results of the laboratory tests of the COVID-19-positive and control group patients are summarized in Table 2. Patients with COVID-19 had substantially greater levels of ACE2, IL-6, TNF-α, glucose, total white blood cells (WBCs), neutrophils, NLR, platelets, mean corpuscular hemoglobin (MCH), and mean corpuscular hemoglobin concentration (MCHC) than the healthy participants. However, their lymphocyte, red blood cell (RBC), hemoglobin, and hematocrit levels showed substantial to high decreases. The difference between eosinophils and mean corpuscular volume (MCV) was not statistically significant.

### 2.2. Serum Levels of Individual Reactive Oxygen and Nitrogen Species

ROS and RNS levels were assessed in the sera of COVID-19 and healthy subjects and the findings are presented in Table 3.

When compared with healthy control participants, the serum concentrations of individual ROS in the COVID-19 patients were significantly elevated. Patients with COVID-19 had elevated levels of hydrogen peroxide, hydroxyl radical, superoxide anion, and singlet oxygen (511.02 ± 218.07 nM, 210.15 ± 68.43 nM, 152.65 ± 62.63 nM, and 129.98 ± 57.74 nM, respectively) compared with those in the control group (377.90 ± 222.40, 151.02 ± 68.71, 109.68 ± 59.06, and 99.40 ± 55.24, respectively).

Individual RNS serum concentrations were considerably greater in the COVID-19 patients than in the healthy controls. Nitric oxide, nitrogen dioxide, and peroxynitrite levels were 27.70 ± 5.91 μM, 5.69 ± 3.34 μM, and 4.56 ± 2.75 μM in the COVID-19 patients, respectively, whereas they were 13.22 ± 3.92 μM, 4.42 ± 1.42 μM, and 3.16 ± 1.31 μM in the healthy control subjects, respectively. Additionally, serum ROS and RNS concentrations were considerably greater in COVID-19 patients admitted to the ICU than in patients with mild or moderate COVID-19 (non-ICU patients) (Figure 1).

### 2.3. Correlations of Individual ROS and RNS Serum Levels with Biochemical Markers in COVID-19 Patients

To correlate the biochemical indicators and serum concentrations of ROS and RNS in the COVID-19 patients, Pearson’s correlation coefficients (r) were computed (Table 4). Moderate to strong positive associations were observed between individual ROS and ACE2 (r = 0.532–0.739) and TNF-α (r = 0.572–0.721), while moderate to very strong positive associations (r = 0.583–0.863) were observed for IL-6. However, the associations with the neutrophil-to-lymphocyte ratio (NLR) were only moderately positive (r = 0.446–0.529).

Regarding RNS, moderate to significant positive associations were observed between serum RNS and IL-6 (r = 0.574–0.659) and ACE2 (r = 0.595–0.638). TNF-α showed weakly positive associations (r = 0.451–0.532), while mild to moderate associations (r = 0.341–0.535) were observed with NLR.

Figure 2 shows a box and whisker plot for the ROS and RNS serum concentrations in the COVID-19 patients and the healthy control group; serum concentrations of ROS and RNS in the COVID-19 patients were considerably greater (*p* ˂ 0.001) than those in the healthy controls.

## 3. Discussion

The production of ROS and RNS is a major mechanism that causes cell death by apoptosis or necrosis, especially in the initial stages of the infection-related immune response [23]. In addition, ROS and RNS are signaling molecules that regulate several physiological systems. They participate in processes that result in the death of viral cells and the restoration of health in patients [24]. RNS are also produced by local pulmonary and inflammatory cells and are involved in the pathogenesis of chronic lung diseases and pulmonary infections [13]. In some cases, COVID-19 infection is associated with ARDS and extensive mortality [25]. Consequently, in severe COVID-19, ROS and RNS may be associated with at least one of the primary disease-modifying pathways. Nonetheless, to the best of our knowledge, ROS and RNS’ sole functionality in COVID-19 and their associations with disease severity have not been thoroughly investigated.

Superoxide anion, hydroxyl radical, hydrogen peroxide, singlet oxygen, and other intermediates of oxygen that are either free radicals or non-free radicals are the ROS involved in COVID-19 pathogenesis. These compounds are generated by NADPH oxidases and other plasma membrane proteins, lipid metabolism, and cytosolic enzyme activity, such as cyclooxygenase activity. Although all these processes contribute to total oxidative stress, the majority of cellular ROS are produced by mitochondria via oxidative phosphorylation [23]. RNS, however, are byproducts of a variety of biological activities, including aerobic metabolism, because L-arginine is converted to the nitric oxide radical by the enzyme nitric oxide synthase, which subsequently interacts with superoxide to generate peroxynitrite [26].

The current investigation evaluated serum concentrations of each reactive species and correlated these findings with COVID-19 biochemical indicators. The hydroxyl radical, superoxide anion, hydrogen peroxide, and singlet oxygen, as well as nitrogen dioxide, nitric oxide, and peroxynitrite were the measured reactive species. The first notable finding was the significant rise in levels of different ROS and RNS in individuals with severe COVID-19 and sepsis in comparison to healthy control patients. This might be explained by the possible participation of the aforementioned reactive species in the COVID-19 inflammatory response. The elevated levels of total ROS in the COVID-19 patient group were consistent with an earlier report; however, the current study measured individual ROS instead of total ROS [24] and RNS [13]. COVID-19 patients who were admitted to the ICU had greater ROS and RNS levels than non-ICU patients. Together, these findings imply that reactive species serum concentration levels upon admission could be an indicator of disease severity and might influence patient survival.

In this study, COVID-19 patients had significantly increased levels of total WBCs, neutrophils, platelets, MCH, and MCHC compared with healthy controls, which is consistent with prior reports [27,28]. This may be due to the impact of the virus on RBC and WBC degeneration. In contrast, the COVID-19 patients had considerably reduced levels of lymphocytes, hemoglobin, RBCs, and hematocrit. Further, key antioxidant enzyme concentrations were lower and protein breakdown markers were higher in the RBCs of COVID-19 patients. Lymphocytopenia and thrombocytopenia were more prevalent in COVID-19 patients admitted to the ICU compared with non-ICU cases. The neutrophil count was significantly increased while the lymphocyte count was significantly decreased in the COVID-19 patient group. Moreover, NLR was increased in COVID-19 patients relative to the controls. The NLR value represents a valuable measure of systemic inflammation and has been investigated as a prognostic biomarker for a variety of chronic disorders [29]. In individuals with severe COVID-19, neutrophils produce a large amount of reactive oxygen species [30]. Herein, somewhat moderate correlations between individual ROS and NLR were observed compared with weak to moderate correlations in the case of individual RNS. This difference may be a consequence of transient lymphopenia, which is dependent on IFN type I and is associated with several viral infections. Additionally, direct viral infection through spike receptors on T cells can cause lymphopenia [31]. These results could assist in assessing the clinical severity of COVID-19 in patients with the disease.

Our study showed higher serum ACE2 concentrations in COVID-19 patients than those in the control group. ACE2 may be able to protect against lung injury by inhibiting the renin–angiotensin–aldosterone system (RAAS) and/or angiotensin II-signaling pathway. Moreover, elevated ACE2 concentrations have been reported to cause increased angiotensin II, which may be an underlying factor of COVID-19-associated ARDS [32]. The hypothesized impact of ACE2 levels on disease severity was validated by the moderate to strong positive correlation between the ACE2 serum levels and ROS (r = 0.532–0.739) and RNS (r = 0.595–0.638) in COVID-19 patients.

Moreover, the serum concentrations of IL-6 in the COVID-19 patient group were considerably higher than those of patients in the control group. A moderate to very strong positive association was observed between IL-6 and individual ROS levels (r = 0.583–0.863), and a moderate to strong positive association was observed between IL-6 and individual RNS levels (r = 0.574–0.659). These findings can be justified by the potential contribution of IL-6 to the activation of human neutrophils and monocytes, which results in an increase in the generation of free oxygen and nitrogen radicals [33].

TNF-α serum levels were noticeably greater in COVID-19 patients than in the healthy group. These high levels revealed moderately positive correlations with individual RNS (r = 0.451–0.532) and moderate to strong positive associations with individual ROS (r = 0.572–0.721). This may be explained by the possible role of TNF-α in triggering mitochondrial ROS generation in association with ATP synthesis [7]. The effect of immunization on ROS levels was documented following vaccination with two doses of mRNA vaccine. It was discovered that a rise in ROS levels was reported after the first dosage, with no changes noted until the day before the second immunization dose [34]. Tissue damage can be caused by excessive inflammation associated with the overproduction of damaging ROS and RNS. Inflammatory cells produce more cytokines in response to the oxidative and nitrative stress brought on by an imbalance between oxidants and antioxidants [8].

Although this study has shown intriguing potential therapeutic associations between serum levels of ROS and RNS and biochemical markers in patients with COVID-19, for the first time, our sample only consisted of patients from one tertiary hospital. Furthermore, the sample was small. Therefore, the conclusions cannot be generalized without the completion of other multicenter trials with larger sample sizes and encompassing other diseases.

## 4. Materials and Methods

### 4.1. Study Population

The current case-control research included 110 COVID-19 patients who were admitted to Ohud Hospital following positive polymerase chain reaction (PCR) test results and were monitored and treated in accordance with the Saudi Ministry of Health-approved protocols. The enrolled patients had ground-glass opacity on computed tomography (CT) scans of their lungs and/or respiratory symptoms that included coughing and shortness of breath. The exclusion criteria included patients with concomitant conditions, such as liver disease, hematological disorders, chronic lung diseases, and those who had received radiation treatment or chemotherapy. Based on negative PCR results and symptoms, 50 healthy control subjects were chosen for study inclusion. The sample size was selected based on the number of COVID-19 patients admitted to Ohud Hospital, approval of the patient to be enrolled in the study, and safety considerations. However, the healthy control subjects were selected from relatives of COVID-19 patients who proved negative COVID-19 test results and did not suffer from any health complications. Unfortunately, only 50 subjects were approved to share in this study. Hence, correction for multiple testing was conducted to adjust *p*-values and to avoid the occurrence of false positives. The research protocol was approved by the General Directorate of Health Affairs in Madinah (No. H-03-M-085). Each subject provided informed consent and data used to identify the subjects were coded and remained confidential.

### 4.2. Sample Collection and Analysis

Venipuncture was performed to obtain blood samples while maintaining aseptic safety standards. Samples were collected from patients just upon admission to the hospital before starting the treatment protocol. Immediately after collection, samples were centrifuged at 2000× *g* for 10 min at 4 °C and then kept at −70 °C until ROS and RNS were individually analyzed. The biochemical markers ACE2, IL-6, and TNF-α were measured in sera of COVID-19 patients and control volunteers. Complete blood counts (CBCs) were performed on the remaining portions of the whole blood samples. Oropharyngeal samples were taken for PCR testing under controlled conditions by qualified specialists from the infection control unit. A virus transfer medium was utilized to transport specimens to the laboratory.

Selective fluorescence-based reagents were used to determine the levels of ROS and RNS in the serum of COVID-19 patients and healthy control volunteers. The Amplex^®^ Red Hydrogen Peroxide/Peroxidase Kit from Molecular Probes (Eugene, OR, USA) was utilized for the precise detection of hydrogen peroxide levels [35]. Serum samples were incubated with Amplex Red at 10 mM for 5 min at room temperature together with 1 U/mL HRP in 50 mM Tris, pH 7.4, and then the reaction was diluted three times before measurement. Excitation at 563 nm and emission at 587 nm were used to assess the fluorescence of the diluted Amplex Red solution in comparison to a freshly prepared H_2_O_2_ reagent solution. The hydroxyl radical was specifically monitored using coumarin-3-carboxylic acid (3-CCA) from Sigma-Aldrich (Seelze, Germany). A 5 mM solution of 3 CCA was prepared in a 20 mM phosphate buffer (PB), pH 7.4, at room temperature. The serum sample was incubated for 2 min with the reagent and the fluorescence of the generated product was measured at emission 450 nm after excitation at 395 nm against the standard prepared using Fenton’s reaction, whereas Fenton’s reaction between hydrogen peroxide and ferrous ammonium sulphate (Sigma-Aldrich) in phosphate buffer (pH = 7.4) was used to produce a hydroxyl radical [36]. By measuring the amount of fluorogenic ethidium (E+) produced by the superoxide anion’s oxidation reaction with hydroethidine (HE) (Sigma-Aldrich), the amount of superoxide anion was quantified. The serum sample was treated with 63.5 mM HE prepared in DMSO and diluted in a phosphate buffer, pH = 7.8. The oxidation of HE to E+ was monitored fluorometrically by exciting at 470 nm and following emission at 590 nm against the standard treated similarly. Potassium superoxide (Sigma-Aldrich) was used as the superoxide anion reference standard [37]. Singlet oxygen was produced as a reference standard for calibration with the Singlet Oxygen Sensor Green (SOSG) from Sigma-Aldrich using the tetra-sulfonated photosensitizing agent porphine tetra (p-phenylsulfonate) (TPPS) from Molecular Probes (Eugene, OR, USA). An amount of 20 µL of a 1 mM solution of SOSG dissolved in CH_3_OH was added to 20 µL of serum sample and then kept in a dark incubator at 37 °C for 2 h. The fluorescence was measured at 536 nm after excitation at 480 nm against the standard [38]. 4,5-Diamino-fluorescein (DAF-2) was utilized as a selective fluorescent reagent to quantify nitric oxide radicals, and spermine nonoate (Sigma-Aldrich) was used as the nitric oxide radical donor. Serum samples were mixed with 100 µL of sodium phosphate buffer and 100 µL of 10 mM DAFs dissolved in DMSO, then kept for about 1 min. Fluorescence was monitored at ex/em 495 nm/515 nm against a standard treated in a similar manner [39]. Nitrite was detected using a 2,3-diaminonaphthalene (DAN) fluorescence kit (Sigma-Aldrich). A serum sample (150 µL) was mixed with 75 µL of 158 µM of DAN solution and 75 µL of 1.5 N of HCl solution, then incubated for 5 min at 30 °C in the dark. A total of 10 µL of 3.0 N NaOH solution was added, and the fluorescence was measured at an excitation of 365 nm and an emission of 410 nm against the standard sodium nitrite [40]. For measurement of nitrate concentration, it was converted into nitrite by the addition of 5 U/mL nitrate reductase (Sigma-Aldrich), and DAN (Sigma-Aldrich) was employed to detect the transformed product. Sodium nitrate and nitrite were employed as calibration reference standards for the nitrate and nitrite radicals, respectively [40]. An enzyme-linked immunosorbent assay (ELISA) was used to measure serum ACE2 levels (ACE2 Kit) Invitrogen from Thermo Fisher Scientific (Waltham, MA, USA). Serum IL-6 and TNF-α levels were measured using an ab46027 ELISA kit from Abcam plc (Cambridge, UK) and Quantikine^®^ ELISA kit from R&D Systems Inc. (Minneapolis, MN, USA), respectively. All ELISA tests were performed exactly as per the manufacturer’s instructions.

### 4.3. Statistical Analysis

The Statistical Package for the Social Sciences from International Business Machines Corp., version 20 (Armonk, NY, USA), was used for all statistical procedures. Student’s unpaired *t*-tests were used to analyze the data. The results are presented as mean values together with their respective standard error. Statistical significance was set as follows: *p* > 0.05 denoting non-significant differences, *p* ≤ 0.05 denoting significant differences, and *p* ≤ 0.001 denoting very significant differences. Evan’s approach was utilized to calculate the correlation strength with Pearson’s correlation coefficient (r) [41], where very strong, strong, moderate, weak, and very weak correlations were considered at r > 0.8, r = 0.6–0.8, r = 0.4–0.6, r = 0.2–0.4, and r < 0.2, respectively.

## 5. Conclusions

The current study assesses the individual ROS and RNS for the first time in association with COVID-19 severity which is more specific, rather than measuring the total oxidative stress. Herein, the pathophysiology of COVID-19 was shown to be significantly associated with various ROS and RNS, although certain species, such as hydroxyl radicals, play dominant roles in disease severity. Moderate to strong relationships between serum ROS and RNS levels and COVID-19 disease severity were elucidated. Furthermore, several inflammatory biomarkers of COVID-19, such as TNF-α, IL-6, ACE2, and NLR, were shown to be highly associated with ROS and RNS concentrations in patient sera. Therefore, ROS and RNS represent potential biomarkers for monitoring the progression of COVID-19. These findings pave the way for identifying rational treatment options for a variety of patient groups with COVID-19 and diseases with similar pathologies.

## Figures and Tables

**Figure 1 ijms-24-08973-f001:**
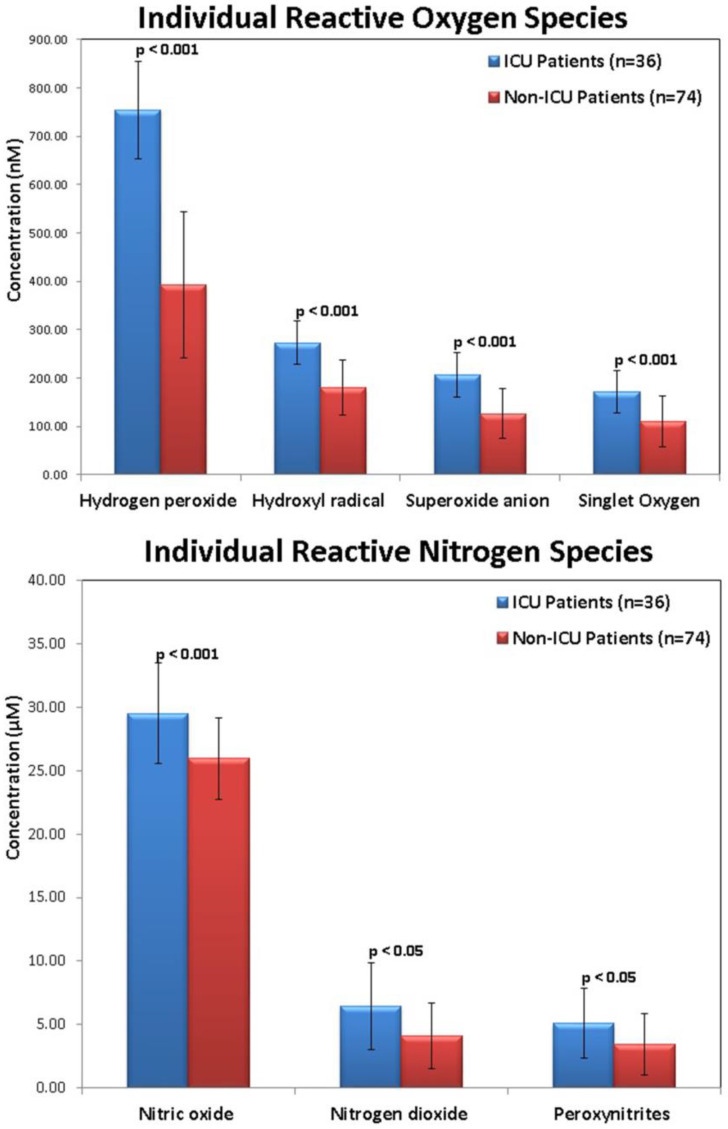
Serum levels of ROS and RNS in ICU COVID-19 patients (*n* = 36) and non-ICU COVID-19 patients (*n* = 74).

**Figure 2 ijms-24-08973-f002:**
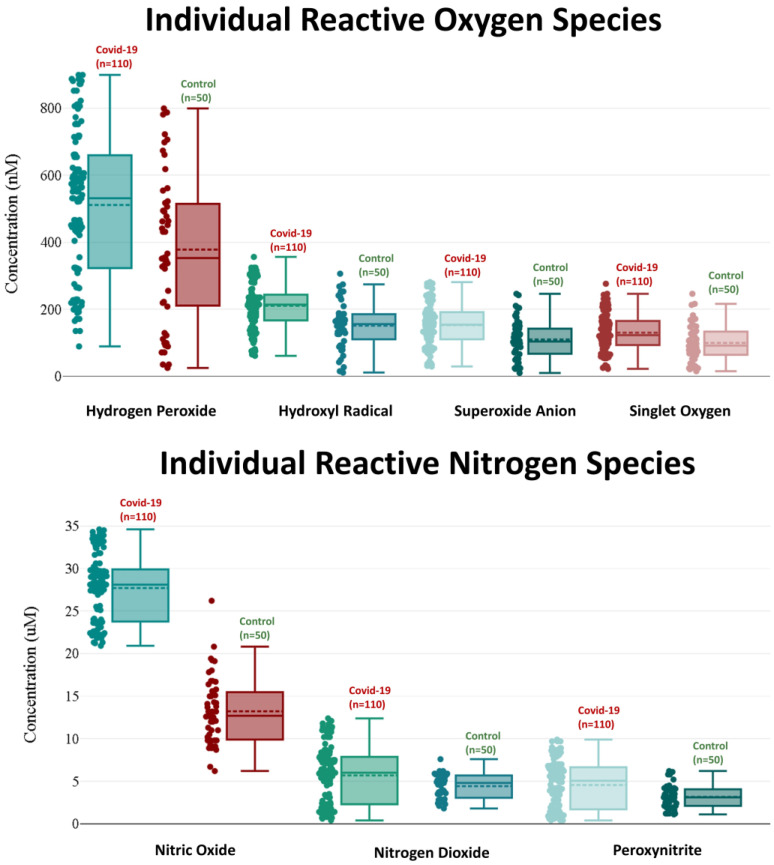
Box and whisker plot of serum levels of ROS and RNS in the COVID-19 patient (*n* = 76) and healthy control (*n* = 34) groups.

**Table 1 ijms-24-08973-t001:** Demographic features of the COVID-19-positive group compared with the control group.

Demographic Features	COVID-19 Group (*n* = 110)	Control Group (*n* = 50)	*p*-Value
Age (years) ^a^	47.92 ± 12.38	49.26 ± 9.94	NS
Sex ^b^			
Male	83 (75.46%)	37 (74.0%)	–
Female	27 (24.54%)	13 (26.0%)	–
Weight (kg) ^a^	73.56 ± 12.26	72.12 ±12.89	NS
Height (cm) ^a^	165.55 ± 9.60	165.06 ± 7.52	NS
BMI (kg/m^2^) ^a^	26.71 ± 3.28	26.47 ± 4.43	NS

BMI, body mass index; NS, non-significant. ^a^ Mean ± standard error of the mean. ^b^ Frequency and ratio.

**Table 2 ijms-24-08973-t002:** Laboratory test results for the COVID-19-positive group compared with the control group.

Test	COVID-19-Positive Group * (*n* = 110)	Control Group * (*n* = 50)	*p*-Value
ACE2 (U/L)	72.01 ± 18.66	32.36 ± 11.27	≤0.001 (HS)
IL-6 (pg/mL)	96.62 ± 25.03	2.76 ± 0.62	≤0.001 (HS)
TNF-α (pg/mL)	7.43 ± 1.93	0.29 ± 0.07	≤0.001 (HS)
Glucose (mg/dL)	187.50 ± 36.83	125.75 ± 25.49	≤0.001 (HS)
Total WBCs (×103/μL)	7.67 ± 2.62	5.18 ± 0.76	≤0.001 (HS)
Neutrophils (%)	57.31 ± 3.13	47.81 ± 6.17	≤0.001 (HS)
Eosinophils (%)	5.10 ± 0.16	5.08 ± 0.17	NS
Lymphocytes (%)	36.57 ± 5.95	45.02 ± 2.64	≤0.001 (HS)
Platelets (×103/μL)	263.11 ± 71.01	215.10 ± 25.38	≤0.001 (HS)
NLR	1.61 ± 0.29	1.07 ± 0.16	≤0.001 (HS)
RBCs (×106/µL)	4.51 ± 0.56	5.09 ± 0.64	≤0.05 (S)
Hb (g/dL)	10.91 ± 1.75	13.25 ± 0.9	≤0.05 (S)
HCT (%)	36.42 ± 4.77	40.92 ± 2.21	≤0.05 (S)
MCV (fL)	83.71 ± 5.30	81.35 ± 3.47	NS
MCH (pg)	30.58 ± 3.47	27.21 ± 2.82	≤0.05 (S)
MCHC (g/dL)	33.95 ± 1.33	30.12 ± 1.75	≤0.05 (S)

* Mean ± standard error of the mean. ACE, angiotensin-converting enzyme; Hb, hemoglobin; HCT, hematocrit; HS, highly significant; IL-6, interleukin-6; MCH, mean corpuscular hemoglobin; MCHC, mean corpuscular hemoglobin concentration; MCV, mean corpuscular volume; NLR, neutrophil-to-lymphocyte ratio; NS, nonsignificant; RBCs, red blood cells; S, significant; TNF-α, tumor necrosis factor-alpha; WBCs, white blood cells.

**Table 3 ijms-24-08973-t003:** Serum levels of reactive oxygen and nitrogen species in the COVID-19-positive group compared with the control group.

Reactive Species Levels	COVID-19-Positive Group * (*n* = 110)	Control Group * (*n* = 50)	*p*-Value
Hydrogen peroxide (nM)	511.02 ± 218.07	377.90 ± 222.40	≤0.001 (HS)
Hydroxyl radical (nM)	210.15 ± 68.43	151.02 ± 68.71	≤0.001 (HS)
Superoxide anion (nM)	152.65 ± 62.63	109.68 ± 59.06	≤0.001 (HS)
Singlet oxygen (nM)	129.98 ± 57.74	99.40 ± 55.24	≤0.001 (HS)
Nitric oxide (µM)	27.70 ± 5.91	13.22 ± 3.92	≤0.001 (HS)
Nitrogen dioxide (µM)	5.69 ± 3.34	4.42 ± 1.42	≤0.001 (HS)
Peroxynitrite (µM)	4.56 ± 2.75	3.16 ± 1.31	≤0.001 (HS)

* Mean ± SEM. HS, highly significant; SEM, standard error of the mean.

**Table 4 ijms-24-08973-t004:** Correlations of laboratory biomarkers in the COVID-19-positive group with serum levels of reactive oxygen and nitrogen species.

Reactive Species	ACE2 (r)	IL-6 (r)	TNF-α (r)	NLR (r)
Hydrogen peroxide	0.739	0.850	0.721	0.529
Hydroxyl radical	0.733	0.863	0.734	0.448
Superoxide anion	0.630	0.715	0.602	0.486
Singlet oxygen	0.532	0.583	0.572	0.446
Nitric oxide	0.638	0.659	0.532	0.535
Nitrogen dioxide	0.552	0.574	0.451	0.461
Peroxynitrite	0.595	0.629	0.501	0.341

ACE, angiotensin-converting enzyme; IL-6, interleukin-6; NLR, neutrophil-to-lymphocyte ratio; r, correlation coefficient; TNF-α, tumor necrosis factor-alpha.

## Data Availability

Data sharing is applicable to this article.

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
