# Peer review of "The Impact of Serum Levels of Reactive Oxygen and Nitrogen Species on the Disease Severity of COVID-19"

_ijms, 2023, doi:10.3390/ijms24108973_

Round 1
Reviewer 1 Report
This manuscript entitled “The impact of serum levels of reactive oxygen and nitrogen 2 species on the disease severity of COVID-19”, by Ahemd et. al, reports the findings of a case-control study of 110 individuals with COVID-19 and 50 healthy controls. The investigators had measured three reactive nitrogen species and four reactive oxygen species and also carried out clinical and routine laboratory of the studied individuals. The investigators had shown that ROS and RNS serum levels were significantly higher in COVID-19 patients compared with control and also noticed a moderate to very strong correlation between the serum levels of the reactive species and the inflammatory laboratory markers. Though the findings could have potential translational impact, I have the following concerns about this study:
1. Have you done correction for multiple testing? Given the multitude of the tests carried out (results reported in Table 2 and Table 3), some of the significant differences noticed could be due to the large number of the statistical tests carried out.
2. What was the rationale for selecting 110 patients with COVID-10 and 50 healthy controls? Did the authors carry out any sample size calculations?
3. Quality of Figure 1 needs to be improved.
4. It is not clear which data belongs to the healthy control and which one belongs to patients with COVID-10 in figure 2.
5. I’m not seeing any patients’ clinical data. Did the investigators assess the association between the clinical factors (e.g., SpO2, PaO2/FiO2, respiratory rate, lung infiltrates percentage, etc.) and the reactive species levels? Merely showing that there is a correlation between reactive species levels and laboratory markers is not a novel observation.
6. Did the authors compare those with severe infection with those with mild infection?
7. When in the disease course did the investigators measured the serum levels of these reactive species? It is important to report the timing in respect to the symptom onset.
8. There are other reports highlighting the importance of reactive oxygen species in those with severe infection (Ref 1) or vaccination (Ref 2). I suggest referring to them in the discussion section of the manuscript and discussing your findings in the context of the previous knowledge on this topic.
9. Reference 13 of the manuscript refers to a publication on COVID-19, I recommend referring to it more specifically in the text rather than discussing it in the general context of pulmonary infections. The authors should then highlight the added value of their study (e.g. higher sample size?) in the context of the role of reactive nitrogen species as biomarkers of COVID-19 disease severity.
10. Data sharing could be applicable to this study since you have data on the clinical, laboratory, and reactive species level for each individual patient. There is no way to extrapolate individual patient data based on the information provided in this paper.
References:
1. 1 Veenith, T., Martin, H., Le Breuilly, M. et al. High generation of reactive oxygen species from neutrophils in patients with severe COVID-19. Sci Rep 12, 10484 (2022). https://doi.org/10.1038/s41598-022-13825-7
2. 2 Lymperaki E, Kazeli K, Tsamesidis I, Nikza P, Poimenidou I, Vagdatli E. A Preliminary Study about the Role of Reactive Oxygen Species and Inflammatory Process after COVID-19 Vaccination and COVID-19 Disease. Clinics and Practice. 2022; 12(4):599-608. https://doi.org/10.3390/clinpract12040063
Moderate editing recommended
Author Response
Response to Reviewer 1 Comments:
Thank you very much for your great efforts we really appreciate to revise our manuscript.
Point 1: Have you done correction for multiple testing? Given the multitude of the tests carried out (results reported in Table 2 and Table 3), some of the significant differences noticed could be due to the large number of the statistical tests carried out.
Response 1: Correction for multiple testing was conducted to adjust p-values and to avoid the occurrence of false positives that may occur due to the difference in sample size between patient group and control group. (The response was added to the revised version of the article section 4.1)
Point 2: What was the rationale for selecting 110 patients with COVID-10 and 50 healthy controls? Did the authors carry out any sample size calculations?
Response 2: The sample size was selected based on the number of COVID-19 patients admitted for Ohud hospital, approval of the patient to be enrolled in the study and safety considerations. However, the healthy control subjects were selected from relatives of COVID-19 patients who proof negative COVID-19 test results and did not suffer from any health complications. Unfortunately, only 50 subjects approved to share in this study. Hence, correction for multiple testing were conducted. (The response was added to the revised version for the article section 4.1)
Point 3: Quality of Figure 1 needs to be improved.
Response 3: Quality of Figure 1 was improved in the revised version.
Point 4: It is not clear which data belongs to the healthy control and which one belongs to patients with COVID-10 in figure 2.
Response 4: The figure 2 was corrected as data for control subjects was missed and all necessary information was added as required in the revised version.
Point 5: I’m not seeing any patients’ clinical data. Did the investigators assess the association between the clinical factors (e.g., SpO2, PaO2/FiO2, respiratory rate, lung infiltrates percentage, etc.) and the reactive species levels? Merely showing that there is a correlation between reactive species levels and laboratory markers is not a novel observation.
Response 5: Concerning clinical factors, they were routinely monitored for patients as most of them were on ventilation according to the disease severity. Hence, the association of these factors with ROS and RNS would give false results. It was better to measure the biochemical factors which reflect the the consequences of cytokine storm.
The current study were concentrated on the biochemical observations in correlation with Individual ROS and Individual RNS for the first time which is more specific. As, the majority of reported research on ROS and RNS in COVID-19 patients has concentrated on measuring overall oxidative stress rather than examining individual ROS and RNS. Even the recommended references by the reviewer assess the overall ROS not individual ROS as in our study.
Point 6: Did the authors compare those with severe infection with those with mild infection?
Response 6: In the current study, serum ROS and RNS concentrations in COVID-19 patients admitted to the ICU (severe infection) were compared to those levels in patients with mild or moderate COVID-19 (non-ICU patients) (Figure 1).
Point 7: When in the disease course did the investigators measured the serum levels of these reactive species? It is important to report the timing in respect to the symptom onset.
Response 7: Samples were collected from patients just upon admission to the hospital before starting the treatment protocol. As the treatment protocol involved the use of antioxidants. ( It was clarified in the revised version of the article)
Point 8: There are other reports highlighting the importance of reactive oxygen species in those with severe infection (Ref 1) or vaccination (Ref 2). I suggest referring to them in the discussion section of the manuscript and discussing your findings in the context of the previous knowledge on this topic.
References:
- Veenith, T., Martin, H., Le Breuilly, M. et al. High generation of reactive oxygen species from neutrophils in patients with severe COVID-19. Sci Rep 12, 10484 (2022). https://doi.org/10.1038/s41598-022-13825-7
- Lymperaki E, Kazeli K, Tsamesidis I, Nikza P, Poimenidou I, Vagdatli E. A Preliminary Study about the Role of Reactive Oxygen Species and Inflammatory Process after COVID-19 Vaccination and COVID-19 Disease. Clinics and Practice. 2022; 12(4):599-608. https://doi.org/10.3390/clinpract12040063
Response 8: These two recommended references were added to discussion section and highlighted in the revised version of the manuscript.
Point 9: Reference 13 of the manuscript refers to a publication on COVID-19, I recommend referring to it more specifically in the text rather than discussing it in the general context of pulmonary infections. The authors should then highlight the added value of their study (e.g. higher sample size?) in the context of the role of reactive nitrogen species as biomarkers of COVID-19 disease severity.
Response 9: The references 13 was necessary in the introduction section in its context. However, the references was highlighted also in the discussion section in the revised version of the manuscript. The major novelty in this article is that the individual ROS and individual RNS were assessed for the first time in association with COVID-19 severity which is more specific rather than measuring the total oxidative stress by all previous reports. We did not consider the sample size high. Hence, it was added as a limitation to carry out multicenter trials with larger sample sizes
Point 10: Data sharing could be applicable to this study since you have data on the clinical, laboratory, and reactive species level for each individual patient. There is no way to extrapolate individual patient data based on the information provided in this paper.
Response 10: Data sharing will be enabled for this study as recommended by the reviewer.
Point 11: Moderate editing recommended
Response 11: The grammar and selling were rechecked and corrected as required.
Reviewer 2 Report
The study investigated the role of ROS and RNS in COVID-19 disease severity. The study is well-designed with appropriate experiments. The results of this study are promising but the presentation needs to be improved. A few aspects below will need to be clarified before it is considered for publication.
The methods need to be written clearly for each of the measurements.
Figure legends should include the number of patients in each analysis.
Figure 2, the number of samples is not on the graphical representation.
Figure 1, please include the number of samples represented in the ROS and RNS graph.
Please confirm each cited reference.
Grammer and spelling checks are required.
Author Response
Response to Reviewer 2 Comments:
Thank you very much for your great efforts we really appreciate to revise our manuscript.
Point 1: The methods need to be written clearly for each of the measurements.
Response 1: The measurement methods for ROS and RNS were were used exactly as reported in the referenced articles and there is no need to repeat again in this article. On the other hand, ELISA tests were used to measure serum ACE2, IL-6 and TNF-α levels exactly as per the manufacturer’s instructions as published on its website.
Point 2: Figure legends should include the number of patients in each analysis.
Response 2: The number of patients in each analysis were added to figure legends the revised version.
Point 3: Figure 2, the number of samples is not on the graphical representation.
Response 3: The number of samples were added to figure 2 in the revised version.
Point 4: Figure 1, please include the number of samples represented in the ROS and RNS graph.
Response 4: The number of samples were added to figure 1 in the revised version.
Point 5: Please confirm each cited reference.
Response 5: All cited references were revised and confirmed.
Point 6: Grammar and spelling checks are required.
Response 6: The grammar and selling were rechecked and corrected as required.
Round 2
Reviewer 1 Report
The manuscript has now significantly improved.
Minor edits
Author Response
Thank you very much for your great efforts we really appreciate to revise our manuscript.
Reviewer 2 Report
The authors have addressed most of the questions but the methods can be improved.
The measurement methods for ROS and RNS were were used exactly as reported in the referenced articles and there is no need to repeat again in this article. On the other hand, ELISA tests were used to measure serum ACE2, IL-6 and TNF-α levels exactly as per the manufacturer’s instructions as published on its website.
The methods described in the manuscript help the reader to use them in their study and cite your paper.
Minor English check.
Author Response
Thank you very much for your great efforts we really appreciate to revise our manuscript.
Point 1: The methods described in the manuscript help the reader to use them in their study and cite your paper.
Response 1: The methods were added to the second revised version of the manuscript according to the reviewer’s advice.